# Understanding the role of policy on inequalities in the intergenerational correlation in health and wages: Evidence from the UK from 1991–2017

Heather Brown  *

Newcastle University, Population Health Sciences Institute, Newcastle upon Tyne, England, United Kingdom

* heather.brown@ncl.ac.uk

## Abstract

Social mobility is high on the policy agenda and is an important component of reducing inequalities. Estimating the relationship across generations of multiple dimensions of mobility such as health and wages can be used to understand the current state of mobility. However, there has been little research on how policy impacts on the relationship of multiple outcomes across generations and how that may be contributing to health inequalities and long-term mobility. In this paper, we use the UK as a case study to evaluate the impact of three distinct policy periods: 1991–1998 (Increasing neo-liberalism); 1998–2009 (English Health Inequalities Strategy); 2010–2017 (Austerity) on the relationship across generations in health (self-assessed health (SAH) and mental health measured by General Health Questionnaire 12 (GHQ-12)) and hourly wages. We employ fixed effects models on data from the British Household Panel Survey (1991–2008) and its successor the Understanding Society Survey (2009–2017). To investigate the role of policy on inequalities, sub-group analysis is performed by parental socioeconomic status measured by parental educational attainment, parental occupation, and if a single parent household. Results show that for the population on average, a changing policy focus has no impact on the strength of the relationship across generations in both health and wages. However, when looking at sub-groups the strength of the relationship in SAH and wages is increasing for parents with basic and higher qualifications and their young adult children. Whereas the influence of parents on their young adult children's SAH, mental health, and wages has remained fairly constant over the period 1991–2017 for parents with manual occupations and professional occupations. There has been a slight weakening in the influence of parents on their young adult children's SAH and wages for single parent families from 2010.

## Introduction

Populist politics is on the rise. Populism is commonly associated with dissatisfaction with systems of power that appear to preserve and entrench class structures. These sentiments have

**Data Availability Statement:** The data underlying the results presented in the study are available from the UK Data Archive.: https://beta. ukdataservice.ac.uk/datacatalogue/studies/study? id=7453. Data cannot be shared publicly because

of governance procedures. The data can be accessed from the UK Data Archive for researchers who meet the criteria for access to the data. Future researchers can register with the UK data service and access the data underlying the study free of charge with the link provided above.

**Funding:** HB-Understanding Society Policy Fellowship https://www.understandingsociety.ac. uk/2018/02/20/understanding-society-policy-fellowships No-The funders had no role in study design, data collection and analysis, decision to public, or preparation of the manuscript.

**Competing interests:** The authors have declared that no competing interests exist.

grown in prominence as welfare regimes in liberal Western democracies have been dismantled [1]. Social mobility is associated with a strong welfare state, lower overall levels of income inequality, and high-quality primary education [2–5]. Intergenerational mobility or lack thereof can cement these feelings of 'being left behind.'

There is a growing body of literature showing that children born into low income families are more likely to end up in poor health as adults [6–8]. A number of studies [8–10] have shown that poor health in childhood impacts on educational attainment leading to lower wages in adulthood. Poor health in adulthood impacts on hours available to work and ability to work which will also have an impact on wages [11–13]. This suggests that health is an important component of social mobility. Investing in health and childhood health in particular may be one mechanism to help improve social mobility and reduce inequalities.

There is a large literature investigating the intergenerational mobility in income and wages. Findings tend to show high-income mobility in Scandinavian countries such as Sweden and lower income mobility in neo-liberal democracies such as the US and the UK [14]. Specifically, Blanden [15] found the intergenerational correlation in earnings in high income countries ranged from 0.15 in Norway to 0.45 in France with a mean of 0.22.

There have been limited attempts to investigate intergenerational mobility in health. Two studies have focused solely on intergenerational physical health mobility. Kim et al. [16] use data from Indonesia finding that if one's father is in poor health this increases the likelihood that his daughter will be in poor health by 29%. Pascual and Cantarero [17] use data on Spain from the European Community Household Panel (ECHP). They find a significant correlation of between 5 to 10% of a parental health effect for those reporting good or excellent health. Johnston et al [18] investigate the intergenerational persistence in mental health across three generations using the British Cohort Survey. They find a correlation in mental health of 0.190. This is reduced to 0.163 after controlling for health at birth, child health at age 5, parental socioeconomic status, and child cognitive status. The correlation in maternal and daughter mental health is approximately 30% stronger than the correlation in maternal and son mental health. This correlation persists across generations with a one standard deviation increase in mental health associated with 0.09 change in their grandchild's mental health.

Halliday et al. [19] is the only paper to look at intergenerational mobility in physical health and income using data from the Panel Study of Income Dynamics (USA). They find an intergenerational correlation in self-assessed health of 0.20–0.25. The cross-generation correlation in income and self-assessed health mobility is 0.25 suggesting that income mobility is largely independent of health mobility. Health mobility is in decline for cohorts born after 1970. Persistence in self-assessed poor health is greater for families where the parents have lower education levels supporting the health-education gradient. Finally, they show that 40% of the intergenerational persistence in health is explained by early life circumstances.

Thus, given the existing literature the mean correlation in wages/earnings is approximately 0.20 which is very similar to the mean correlation in health. However, it is important to note that there is a lot of variation across this correlation and some of this variation is country dependant. These similarities in the literature of the correlation in two distinct outcomes: health and wages across generations are one of the motivators for this study.

Estimating correlations in multiple outcomes, such health and earnings between generations are an important component of understanding mobility. Mobility is multidimensional and should not be limited to a single domain such as earnings. Thus, in order to create effective policy to ensure equity of opportunity for all it is necessary to understand how policy may impact on multiple dimensions of mobility such as health and wages. For the development of future policy, it is essential to understand how the strength of the correlation across these multiple outcomes is changing over time. As different dimensions may require a bespoke policy

solution if the strength of the correlations in outcomes are moving in opposite directions. If government policy can influence the strength of the intergenerational correlation across multiple dimensions of mobility then this would be one key mechanism to influence mobility in the medium/long term.

In the UK, from the period of 1979–1997, the government pursued a policy of deregulation, which included reducing the power of unions, and welfare reform to reduce the role of the state in people's lives. Over this period, child poverty rates trebled, and income and wealth inequality rose [20]. The Labour government which came into power in 1997 put reducing health inequalities and child and pensioner poverty at the heart of their policy agenda, with its Health Inequalities Strategy (1999–2010). It was the first country in Europe to put in place a systematic policy to reduce socioeconomically driven health inequalities [21]. The total budget of the programme was £20 billion and focused on reducing child and pensioner poverty, geographically related differences in life expectancy, socioeconomically driven infant mortality rates by 10%, Sure Start (early life), smoking cessation services, primary care in inner cities, and better access to treatment for cardiovascular disease and cancer. Evaluation of the Health Inequalities strategy suggests that it was successful at reducing child poverty and the expansion of Sure Start in reducing inequalities in education [22]. Recent studies [23,24] have shown that the English health inequalities strategy reduced geographical related health inequalities in life expectancy and infant mortality rates respectively. In 2010, the Coalition government introduced a period of government austerity retrenching most of the investment in the health inequalities strategy which has continued to the present day. Robinson et al. [23] shows that this change in policy focus has led to a rise in absolute inequalities in infant mortality in England.

The distinct changes of political focus over this period make the UK an excellent case study to understand if and how and for whom government policy has impacted on the strength of the intergenerational correlation in self-assessed health (SAH), mental health measured by the General Health Questionnaire 12 (GHQ-12), and wages. We use data from the British Household Panel Survey (BHPS) and Understanding Society Survey (USS) over the period 1991–2017 to investigate this; splitting the data into 3 distinct policy periods: 1991–1999; 1999–2010; and 2010–2017. We investigate differences in the strength of the correlation across the three outcome variables between generations over these 3 policy periods by socioeconomic status measured by parental education, parental occupation, and single parent versus two parent households. This is a descriptive analysis to understand how family mobility is associated with the policy environment. To date there has been no research on the strength of correlations of multiple outcomes between generations; how these have evolved as a response to the changing political landscape and what this means for inequalities in long-term mobility.

## Theoretical framework

### Health and wages

In the analysis, we choose to look at two outcomes: health and wages which we assume are key components of mobility. The theoretical decision behind this choice is the Grossman [24] framework that health is a consumption and production commodity. Individuals invest in their health as they derive utility from good health and being in good health means that individuals can invest time into other activities such as work to obtain income for consumption of other goods and looking after children. Health is a function of the initial health stock (which is determined by genetics), investments in health such as medical care and preventative care such as exercise and a healthy diet, and the wider environment (social determinants of health) such as housing stock, access to green space, access to health care, sense of belonging,

socialising with family and friends, employment opportunities, and job quality. Individuals in good health are in a better position to take advantage of employment opportunities, obtain higher quality employment and be more productive at work (compared to someone in poor health) resulting in a higher probability of higher wages everything else being equal. Individuals in good health will have more time than those in poor health to invest (both health and their wider development) in their children. These children will then have a higher likelihood of being in good health with the resources and opportunities to invest in their health and economic prospects. Conversely, those in poor health will have less time, economic opportunities, and resources resulting in lower wages and reduced investment in their children. These children then have a higher likelihood of being in poor health themselves and reduced opportunities for high quality employment resulting in lower wages. This is reflected in Fig 1 outlining how poor health can lead to multigenerational poor health and low wages.

Government policy impacts on individual health. Evidence suggests that over the period (1979–1997) child poverty rates trebled [25] and between 1975–1998 the number of people reporting a limited long standing illness increased from 15% to 22% and the number of people reporting an illness in the past two weeks nearly doubled from 9% to 16% [26]. During the

**Fig 1. The poverty and poor health trap.**

Labour's government's English Health Inequality Strategy (1999–2010) [27] there was improved health for those in the lowest socioeconomic groups measured by reductions in mortality (both adult and infant) and a reduction in child poverty [22,23]. A retrenchment of this policy following the 2008 Global Financial Crisis has led to rising child poverty rates and rising infant mortality rates [23].

The correlation in parents and their young adult offspring health is determined by genetics, past parental investments and the wider determinants both past and current. At the individual level, we assume health will influence potential wages. This would mean at the family level or across generations, we would expect that families with better health are more likely to be families that also have higher wages. Thus, without government intervention there would be a polarisation in outcomes. There would be a high correlation between parents and children in both health and wages for high socioeconomic groups where parents have more resources and are better able to access preventative services and medical care. There would be a high correlation between parents and children in both health and wages at the lowest end of the socioeconomic spectrum where parents have less financial resources, restricted access to services, and potentially worse health reducing the time they have available for their children. For those in middle, a priori it is difficult to predict the strength of the intergenerational correlation across health and wages. There may be some upward and downward mobility reducing the correlation in parental and offspring outcomes depending upon parents' health and employment, and job quality. It is also not clear if the correlation in health and wages between generations will be moving in the same direction or if they will diverge.

Government intervention such as the minimum wage and improving access and increasing the availability of health services and the converse of rationing services will impact on both health and wages. It is likely that these policies will have a greater impact at the lower end of the socioeconomic spectrum. However, policies may not have the same impact on both outcomes. Thus, the impact of policy on the relationship between generations in both health and wages is complex. Firstly, there is likely to be a lag of policy on outcomes at the family level. Policies at critical points in child development such as the first five years of a child's life may have long-term consequences on health and wages. Thus, when we look at changes in the strength of the relationship in health and wages over time across generations, we are assessing not only the current economic climate but past policy which will have long term impacts on health and wage potential. Thus, in the econometric model we aim to address the following questions:

1. Did a systematic focus on reducing child poverty and health inequalities over the period 1999–2010 impact on the strength of the relationship between generations in both health and wages? Was the impact same or different for the two outcomes of health and wages?

2. Is there a differential impact of policy focus on the strength of the relationship between generations in health and wages across the socioeconomic spectrum as measured by parental educational attainment, social class, and for single parent families compared with two parent families?

## Methodology

### Data

The data we use comes from the British Household Panel Survey (BHPS) which was administered from 1991–2008 and its successor survey Understanding Society Survey (USS) 2009–2016 [28].

The BHPS was a household survey which began with 5,500 household consisting of 10,300 individuals across 250 areas of Great Britain. Additional samples of 1,500 households in Scotland and Wales were added to the main sample in 1999. In 2001, 2000 households from Northern Ireland were added. In the final wave of the survey participants were asked if they would be interested in joining the new USS. Approximately 6,700 of the 8000 respondents in the final wave of the survey joined USS. They were included in USS from wave 2 (2010) [28].

The USS is the largest household study of its kind consisting of 40,000 households. The study is powered to explore differences between the four countries of the UK and some ethnic minority groups [28].

Both studies are drawn from a random sample of households. A two-stage sampling procedure was used. Primary selection was based upon postcode which were then grouped into geographical strata to ensure a nationally representative selection of households. For more information on study design see [29]. All household members 16 and older are interviewed annually in both surveys. If original household members such as adult children form new households, they continue to participate in the survey. The survey asks respondents a range of questions related to their health, labour market experience, finances, opinions, family life, and well-being.

In the BHPS dataset, we have 22,800 observations on 3100 offspring between the ages of 16–74 and their parents (n = 2051) over the period 1991–2008. In the USS dataset, we have 23,764 observations on 5293 offspring and their parents (n = 3249 parents) over the period 2009–2016. In total we have data for 25 years on different families.

Estimating the intergenerational correlation in health and income is challenging. Firstly, data is required on adults from two generations preferably at multiple time points. Health cannot be directly observed which means that it is prone to measurement error from self-reporting bias. There are also difficulties with accurately measuring the correlation in wages of parents and their children. Wages are likely to suffer from both self-reporting bias and the possibility of transitory fluctuations [14]. Transitory fluctuations are likely to be a bigger issue for children who are at the start of their career and more likely to change jobs regularly.

Typically, to overcome these issues, it is suggested to use multiple time points across the life cycle for both generations to generate an average variable to proxy for lifetime wages and/or health [14]. This will then create a time constant latent variable for health and income. However, it is not clear what the appropriate number of data points required to create an accurate representation of lifetime health and wages; which may bias the findings.

To overcome, these biases when estimating the intergenerational correlation in health and wages we employ a new approach. We use the mean correlation for the sample estimated for each of the 25 years of data so we can look at variation in the relationship between generations over time. Thus, we are not estimating correlations for individual parent and children but the average for the sample and average correlations for the socioeconomic characteristics of interest. Using the population mean for the correlation between generations, means that we remove the impact of individual family genetics on the correlation. This approach will also remove the impact of transitory fluctuations at the family level. In addition, if the survey, remains representative of the overall population; using the population correlation means that we do not need to be concerned with sample attrition impacting on our findings. As the sample, is young adults and their parents we expect the correlations to be a lower bound estimate of the entire population correlation as children and parents are at different points in their life cycle.

## Outcome variables

In the analysis, we are interested in the relationship between generations in both health and wages. We define generations as parents and their young adult children (aged 16+). Parents

and children are identified in the dataset using the study id of the mother and/or father. This id is available in the data for all children of the parents irrespective of the child's age.

The key outcomes we are interested in are:

1. The mean correlation in self-assessed health (SAH) between generations. This is measured by a question asking respondents how they rate their health compared to others their own age.

2. The mean correlation between generations in mental health measured by the GHQ-12-a 36 point scale to identify poor psychological health in a general population.

3. The mean correlation between generations in log of hourly wages. Logs are used to normalise the wage variable. The wages have been adjusted for inflation with 2016 as the base year.

## Key explanatory variable

Our key explanatory variable is the policy period. This is derived from the survey year and is split into 3 distinct periods: 1991–1999 (Increasing Neo-Liberalism); 1999–2010 (Health Inequalities Strategy) and 2010–2017 (Austerity). This variable is created using marginal spline terms, which takes account of time trends with break points to separate the three policy periods.

## Estimation model

The analysis focuses on estimating if the 3 time periods (1991–1999; 1999–2010; 2010–2017) was significantly associated with changes in the strength of the intergenerational correlation in health and wages. To understand how government policy over these time periods may have contributed to changes in the strength of the correlation by socioeconomic status, we interact time with parental educational attainment, parental occupation, and a dummy for if it is a single or two parent household. We employ a fixed effects (FE) segmented linear regression model, measuring the time period using marginal spline terms with a break point in 1999 and 2010 to account for the three policy periods.

Formally the estimation model is:

$$Mean\ Intergenerational\ Correlation_t = \alpha_0 + \beta_1 t_1 + \beta_2 t_2 + \beta_3 t_3 + \mu_p + \varepsilon_{f,t} \tag{1}$$

Where Mean Intergenerational Correlation is the mean of the individual family correlations for the sample and sub-samples in SAH, GHQ, and wages respectively and are estimated in separate models. The subscript $t$ represent each year over the period 1991–2017. $\alpha_0$ is the constant term. $t_1, t_2$, and $t_3$ are the marginal spline terms representing the time trends over the 3 periods with the associated parameter of coefficients to be estimated which are represented by the $\beta$ terms. $\mu$ is a time constant error term for the population, $p$. This is removed from the model with the fixed effects approach. Fixed effects are used to reduce bias from time constant factors related to the wider landscape impacting on the correlation between generations. As we are not looking at specific elements of policy but the wider policy landscape, a more general-ised model such as a fixed effect is more appropriate than a quasi-experimental model such as difference-in-difference. We do not have an obvious 'treatment effect' or control group. With this approach, we can draw some conclusions on the relationship between the policy environment and mobility, but we don't necessarily know what is driving this relationship. In some model specifications the marginal spline terms are interacted with parental educational attain-ment, parental occupation, and a dummy for single parent household respectively.

## Robustness tests

To test the robustness of our findings we estimated models using different break points to our policy period to ensure that the findings are not being driven by a third factor rather than the policy period. We also assess our choice of using a fixed effects framework by employing random effects models and estimating a Hausman test to determine the most appropriate model specification.

## Results

The first step in the analysis is to check our sample to ensure that the composition of individuals within the sample are similar across the 3 time periods. Over the three time period, we want the mean characteristics of parents and children to be similar so that we know that any change in the correlation is attributable to other factors such as the policy environment rather than changing family composition. Table 1 shows some sample means and standard deviations for the sample over the 3 time periods. The age of offspring over the period has remained constant. Parents have gotten slightly older from an average age of 49 over 1991–1998 to 52 over 2010–2017. Over time there are also more female offspring in the sample moving from 43% in 1991–1998 to 50% in 2010–2017. Both young people and their parents are more likely to report being in good/excellent health over time. Mental health for both young people and parents has remained relatively constant over the sample period. The log of hourly wage has increased for both young adults and their children. There has been a sharp reduction in the percentage of parents with basic qualifications over the sample period going from 74% in 1991–1998 to 32% in 2010–2017. However, the percentage of parents with degree level qualifications has remained relatively constant over the sample period. The percentage of parents in routine/manual occupations has remained at approximately 23% but a higher percentage of parents are working in professional occupations over time (41 to 48%). There has been some fluctuation over the sample period in the number of single parent households. The sample has changed slightly over time, but this reflects wider changes to society over this time period such as changes in educational attainment and occupation profiles (see for example [30]). Thus, our

**Table 1. Descriptive statistics.**

|                          | 1991–1998     | 1999–2009     | 2010–2017     |
|--------------------------|---------------|---------------|---------------|
| Age (Offspring)          | 22.13 (7.81)  | 22.97 (8.62)  | 22.84 (8.33)  |
| Age (Parents)            | 49.83 (9.60)  | 51.04 (10.18) | 52.11 (9.54)  |
| Gender (Offspring)       | 0.43 (0.50)   | 0.48 (0.50)   | 0.50 (0.50)   |
| SAH (Offspring)          | 0.78 (0.41)   | 0.81 (0.40)   | 0.90 (0.31)   |
| SAH (Parents)            | 0.69 (0.40)   | 0.67 (0.42)   | 0.74 (0.40)   |
| GHQ (Offspring)          | 25.98 (5.03)  | 25.74 (5.44)  | 25.63 (5.29)  |
| GHQ (Parents)            | 24.16 (4.88)  | 23.87 (5.32)  | 24.09 (5.33)  |
| LHW (Offspring)          | 1.41 (0.50)   | 1.78 (0.49)   | 1.98 (0.56)   |
| LHW (Parents)            | 1.78 (0.47)   | 2.14 (0.47)   | 2.43 (0.51)   |
| Parents Basic Qual.      | 0.74 (0.44)   | 0.62 (0.49)   | 0.32 (0.47)   |
| Parents Degree           | 0.38 (0.48)   | 0.45 (0.50)   | 0.39 (0.49)   |
| Parents manual/routine   | 0.23 (0.42)   | 0.24 (0.43)   | 0.23 (0.42)   |
| Parents Professional     | 0.41 (0.49)   | 0.42 (0.49)   | 0.48 (0.50)   |
| Single Parent            | 0.38 (0.48)   | 0.43 (0.49)   | 0.32 (0.47)   |

Percentages shown except for Age, GHQ, and log of hourly wage. For ease of interpretation SAH is presented as a binary variable where the percentage shown is those reporting good/excellent health.

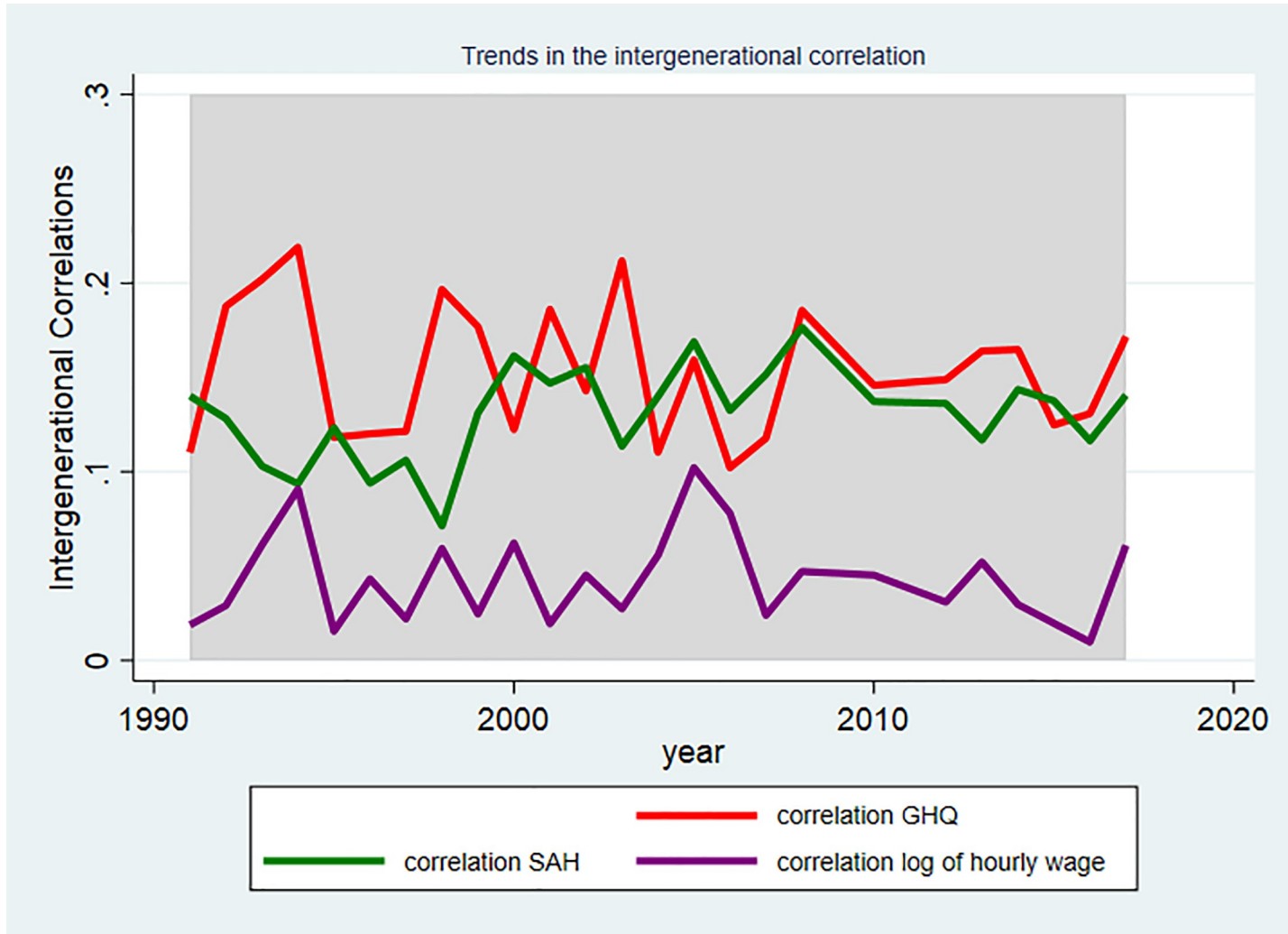

**Fig 2. Trends in the intergenerational correlation in SAH, GHQ, and log of hourly wages between 1991–2017.**

sample should be able to identify how changing political focus related to families and inequalities has impacted on the intergenerational correlation in both health and wages.

Trends in the intergenerational correlation for the entire sample are presented in Fig 2. The mean correlation in the whole sample starts in 1991 at approximately 0.05 for wages, 0.10 for SAH and 0.12 for GHQ. As would be expected because of the age of offspring in the sample, the correlation in log of hourly wages is lower than the correlation in SAH and mental health in Fig 1. There is no obvious difference over the 3 policy periods evident in Fig 1.

The sub-groups by educational attainment (Fig 3A, 3B and 3C), parental occupation (Fig 4A, 4B and 4C), and single vs two parent families (Figs 5A, 5B and 5C) are shown separately by outcome. In Figs 3A and 5C for the sub-groups there is some variation in the correlations between generations. There is not a significant correlation in wages for highly educated parents and their young adult children, single parent families and their young adult children. Whereas there is a correlation of 0.20 for parents in routine/manual profession and their children in 1991. As would be expected since the young adult children are at a different point in the life cycle than their parents these correlations are smaller than those found in the literature

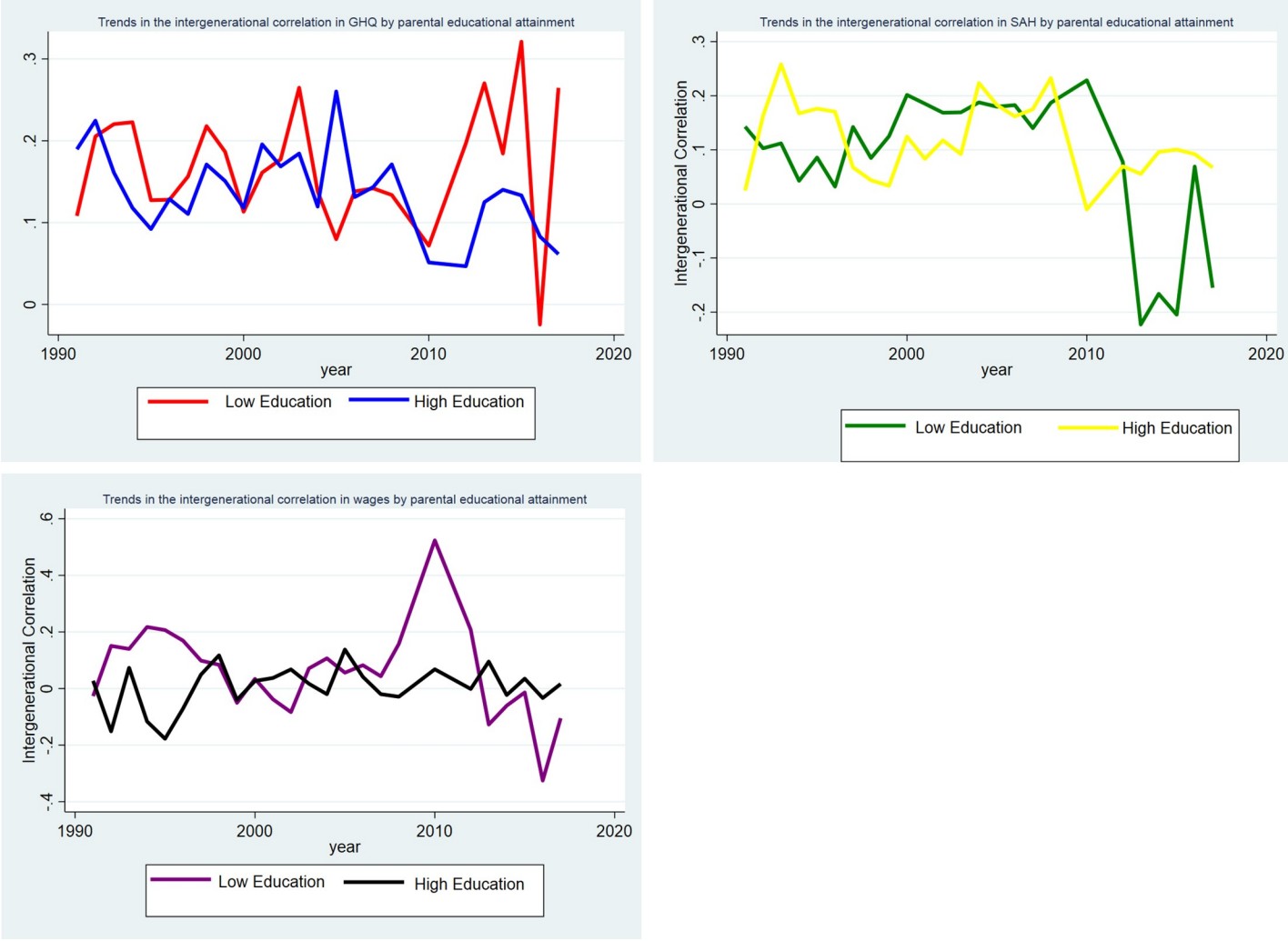

**Fig 3.  A:** Trends in the intergenerational correlation in GHQ between 1991–2017 by parental educational attainment. **B:** Trends in the intergenerational correlation in SAH between 1991–2017 by parental educational attainment. **C:** Trends in the intergenerational correlation in log of hourly wages between 1991–2017 by parental educational attainment.

[14–19]. In the analysis, we are interested in if the influence of parents on their children's outcome is increasing or decreasing over time in response to a changing policy environment.

In Fig 3A, 3B and 3C the gap between the highest and lowest and the different outcome measures appears to be widening from 2010. There also appears to be a widening gap in SAH by parental occupation in Fig 4B. There is less of a clear pattern for the other outcomes for parental occupation. For single vs two parent households there appears to be a convergence across all three outcomes in Fig 5A, 5B and 5C from 2010–2017.

Across all model specifications for the main and sub-group analysis, a Hausman test indicated that the fixed effects model was the preferred specification over a random effects model.

Table 2 shows how the intergenerational correlation in SAH, mental health, and hourly wage evolved over the period 1991–2017. The first thing to note is that the strength of the correlation did not evolve in the same way for SAH, mental health, and wages. For SAH, the correlation strengthened between 1991–2010 and then weakened between 2010–2017. Overall, these two effects cancelled each other out so overall the correlation has not changed over time

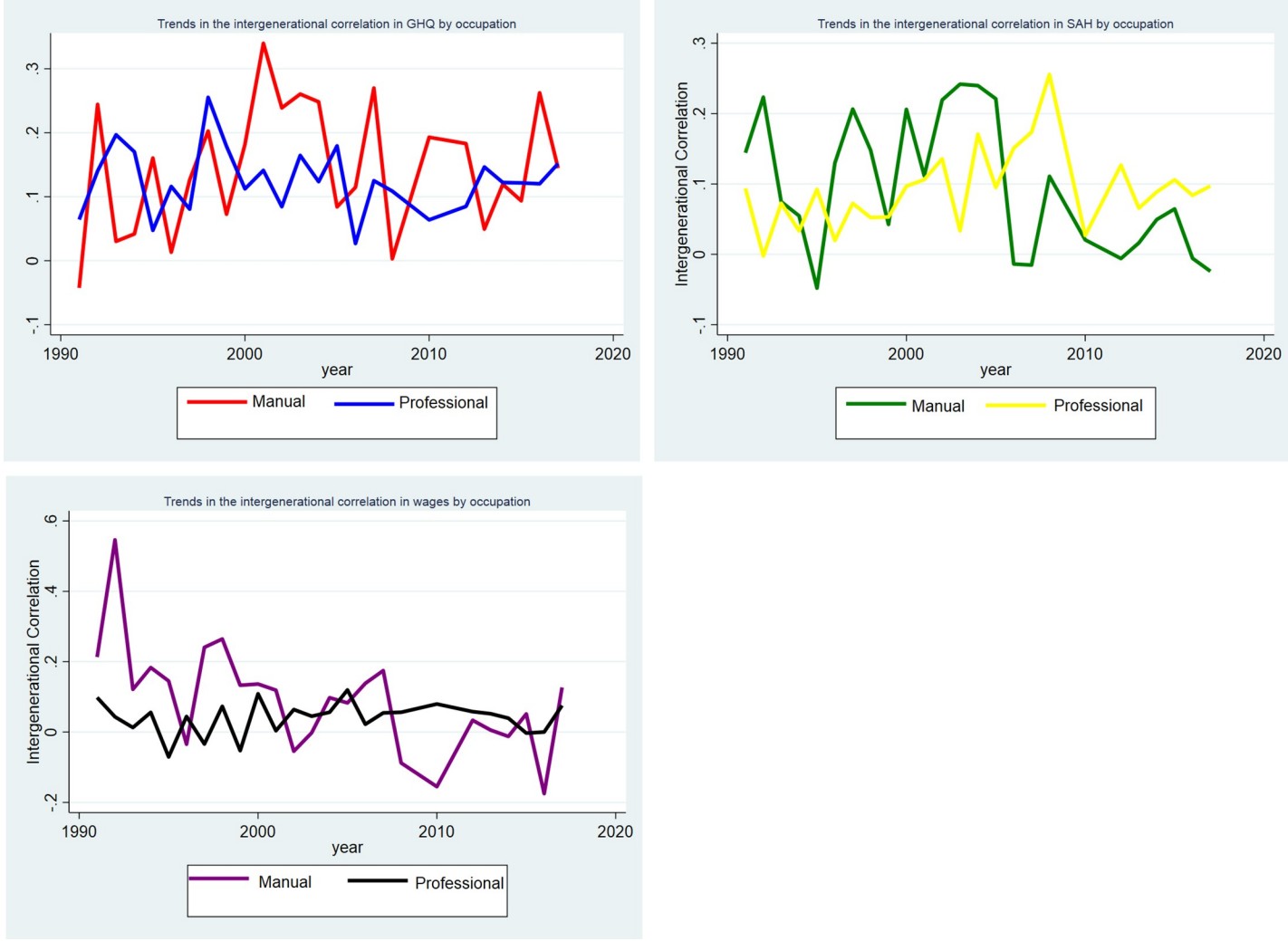

**Fig 4. A:** Trends in the intergenerational correlation in GHQ between 1991–2017 by parental occupation. **B:** Trends in the intergenerational correlation in SAH between 1991–2017 by parental occupation. **C:** Trends in the intergenerational correlation in log of hourly wages between 1991–2017 by parental occupation.

for SAH. A similar overall effect is found with GHQ, leading to no overall change in the intergenerational correlation over this period. Whereas for wages, the decline in the intergenerational correlation from 2010–2017 was larger than any gains from previous period resulting in a small decline in the intergenerational correlation in wages.

Next, looking at the sub-group analysis in Tables 3, 4 and 5. The changes in the strength of the intergenerational correlation are not consistent across different ways of measuring socioeconomic status. In Table 3, the intergenerational correlation in SAH is strengthening for both those with parents with basic qualifications and degree level qualifications over time with this increasing over the 2010–2017 period. Whereas overall there is no real change in the strength of the intergenerational correlation for GHQ. There has been a large increase in the strength of the correlation in wages for those with parents with basic and university qualifications over the period 2010–2017. This helps to quantify the trends observed in Fig 2A which shows some evidence of widening inequalities. In Table 4, overall the intergenerational correlation in SAH, mental health, and hourly wage has remained fairly constant for those with parents in routine/

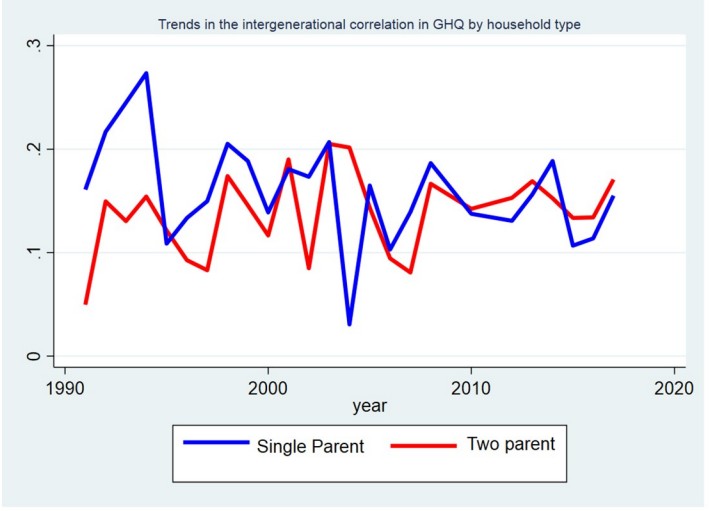

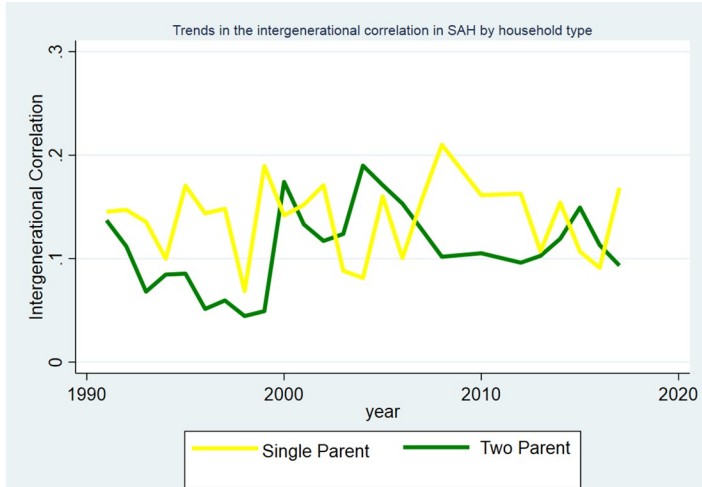

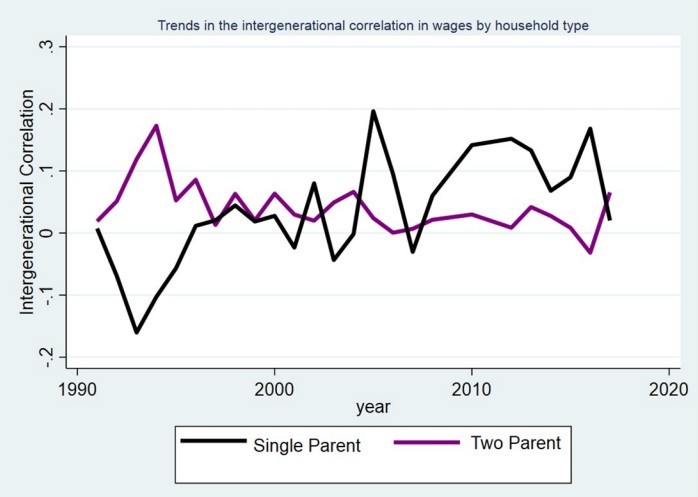

**Fig 5. A:** Trends in the intergenerational correlation in GHQ between 1991–2017 by single vs two parent household. **B:** Trends in the intergenerational correlation in SAH between 1991–2017 by single vs two parent household. **C:** Trends in the intergenerational correlation in log of hourly wages between 1991–2017 by single vs two parent household.

manual occupations and professional occupations. In Table 5, the intergenerational correlation in SAH and hourly wage has slightly weakened over the 2010–2017 period whereas there has been no real change in the correlation in GHQ.

**Table 2. Trends in the intergenerational correlation in SAH, GHQ, and log of hourly wage for the entire sample over the period 1991–2017.**

| Time period | SAH | GHQ | Log of Hourly wage |
|---|---|---|---|
| 1991–1998 | 0.001*** (0.0001) | 0.0003 (0.0002) | 0.0002 (0.0002) |
| 1999–2009 | 0.002*** (0.0002) | -0.002*** (0.0002) | 0.001**0.0002) |
| 2010–2017 | -0.004*** (0.0001) | 0.002** (0.0002) | -0.003*** (0.0002) |

*** indicates p<0.001

**indicates p<0.05

**Table 3. Trends in the intergenerational correlation in SAH, GHQ and Log of hourly wage by parental educational attainment over the period 1991–2017.**

| Time period | SAH | | GHQ | | Log of Hourly wage | |
|---|---|---|---|---|---|---|
| | Basic Qual | Degree | Basic Qual | Degree | Basic Qual | Degree |
| 1991–1998 | -0.003***(0.001) | -0.02*** (0.001) | -0.002** (0.0001) | -0.002** (0.0001) | 0.03*** (0.002) | 0.03*** (0.002) |
| 1999–2009 | 0.0004 (0.001) | 0.03*** (0.001) | 0.01** *(0.001) | 0.01** *(0.001) | -0.06*** (0.002) | -0.04*** (0.002) |
| 2010–2017 | 0.03***(0.002) | 0.04*** (0.002) | -0.01** *(0.002) | -0.01** *(0.002) | 0.13*** (0.004) | 0.09*** (0.004) |

*** indicates $p<0.001$

**indicates $p<0.05$. Coefficient is time period interacted with parental educational attainment.

## Discussion and conclusion

We found that for the population on average, changing priorities by the Government on reducing health inequalities did not have an impact on family mobility. However, when looking at sub-groups in the population as defined by different ways of measuring socioeconomic status, the picture is more heterogeneous. The correlation in SAH and wages is strengthening for parents with basic and higher qualifications. Whereas the correlation in SAH, mental health, and wages has remained fairly constant over the period 1991–2017 when comparing parents with a manual occupation and professional occupation. There has been a slight weakening in the correlation in SAH and wages for single parent families compared to two parent families from 2010. These findings suggest that although we observe in the literature [14–19], similar correlations between generations in health and wages; the policy environment may have differential effects on the influence of the family on health and wages depending upon parents' socioeconomic status. This is important going forward, as some policies may inadvertently increase inequalities if they do not have the same effect on all types of families. Inequality assessments are important when developing new policies aimed at improving childhood outcomes and/or mobility.

Reducing inequalities and supporting social mobility is high on the policy agenda. Existing evidence [23,24] showed that the English Health Inequalities Strategy reduced inequalities in life expectancy and infant mortality which then began to rise again post-policy period. This was the first study to look at how the English Health Inequalities Strategy impacted on inequalities within and between families as measured by the correlation in SAH, mental health, and wages. At the individual level, health and wealth are highly correlated. At the family level, we do not find evidence that the correlation between health and wages is moving in the same direction. This is important to consider when developing policies to improve mobility as there may not be the same spillovers between health and wealth that we observe at the individual level.

Results from studies at the individual level such as [23,24] found a large impact of the health inequality strategy period on geographical health inequalities and infant mortality respectively.

**Table 4. Trends in the intergenerational correlation in SAH, GHQ and log of hourly wage by parental occupation over the period 1991–2017.**

| Time period | SAH | | GHQ | | Log of Hourly wage | |
|---|---|---|---|---|---|---|
| | Routine/Manual | Professional | Routine/Manual | Professional | Routine/Manual | Professional |
| 1991–1998 | -0.01*** (0.001) | -0.01*** (0.001) | -0.02*** (0.001) | -0.01*** (0.001) | 0.02*** (0.001) | 0.02*** (0.001) |
| 1999–2009 | 0.02*** (0.001) | 0.02*** (0.001) | 0.03*** (0.001) | 0.01*** (0.001) | -0.01*** (0.002) | -0.003 (0.002) |
| 2010–2017 | -0.01*** (0.001) | -0.01*** (0.001) | -0.01*** (0.001) | 0.01*** (0.001) | -0.02*** (0.001) | -0.03*** (0.002) |

*** indicates $p<0.001$, **indicates $p<0.05$. Coefficient is time period interacted with parental occupation (intermediate professions are the base category).

**Table 5. Trends in the intergenerational correlation in SAH, GHQ and log of hourly wage by single parent household over the period 1991–2017.**

| Time period | SAH | GHQ | Log of Hourly wage |
|---|---|---|---|
| 1991–1998 | -0.0002 (0.0003) | -0.002*** (0.0004) | 0.02*** (0.001) |
| 1999–2009 | -0.0001 (0.0004) | 0.01*** (0.001) | -0.004*** (0.001) |
| 2010–2017 | -0.004*** (0.0004) | 0.0003 (0.0005) | -0.02** (0.001) |

*** indicates p<0.001

**indicates p<0.05. Coefficient is time period interacted with parental occupation (two parent families are the base category).

Where we found less evidence that policy period changed the influence of parents on their young adult children's outcomes. However, there is some evidence from the sub-group analysis to support the idea that inequalities for some families may be widening since 2010. Thus, the policy environment may have some impact on the role of the family on the life chances of young people. Similar to the findings from Halliday et al. [19] that health mobility is independent of wage mobility, we find that the correlation in SAH, mental health, and wages do not move in the same direction. This suggests that mechanisms explaining the correlation between health and wages at the individual level may not be playing out at the family level. In addition, the policy environment may be increasing/decreasing the influence of the family on health and/or wages which may impact on long term individual health and economic outcomes and subsequently inequalities.

Future research should estimate models following the same families over time, to understand how policy impacts on the role of the family on long term mobility. By estimating correlations for young adults and their parents, we may be missing some of the nuances of government policy on the role of mobility as in our sample parents and their young adult children are at different life stages. This paper is descriptive and does not specifically explore policy levers or mechanisms which may influence the role of the family on long term mobility. Future work should explore specific policies to identify causal relationships between policy and mobility. As we are using an unbalanced panel, sample attrition may impact on the generalisability of our findings to the UK population. There may also be measurement error with our outcome variables as they are self-reported. Even if this measurement error is systematic by our SES indicators, if it moves in the same direction for parents and their children this should minimise the bias in our findings.

Our findings provide some evidence that the influence of parents on their young adult's health and wages does evolve over time and may be responsive to changes in policy focus especially for those at the highest and lowest end of the socioeconomic spectrum. Changing influence of parents' on their children's health and wages may be one factor contributing to the rise and fall in inequalities.

## Acknowledgments

This research was funded by an Understanding Society Policy Fellowship.

## Author Contributions

**Formal analysis:** Heather Brown.

**Project administration:** Heather Brown.

**Software:** Heather Brown.

**Writing – original draft:** Heather Brown.

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
