## [Decision Letter · Decision Letter 0]

28 Jan 2020

PONE-D-19-32818

Understanding the role of policy on inequalities in the intergenerational correlation in health and wages: Evidence from the UK from 1991-2017.

PLOS ONE

Dear Dr Brown,

Thank you for submitting your manuscript to PLOS ONE. After careful consideration, we feel that it has merit but does not fully meet PLOS ONE’s publication criteria as it currently stands. Therefore, we invite you to submit a revised version of the manuscript that addresses the points raised during the review process.

Your paper was sent to over two dozen reviewers; however, only one scholar accepted the invitation to review your work.  As a result, I am offering the reviewer's comments and my own review, per Plos One editorial guidelines and practices, in lieu of a second external review.  

In addition to Reviewer 1's comments and questions, I request that your revision resolve several minor issues.  First, and importantly, you should define SAH in the abstract and earlier in the paper.  You use the acronym repeatedly before defining the concept on page 7.  Please use acronyms only after the the concepts and datasets have been spelled out and explained.  Along these lines, please change the acronym for the Understanding Society Survey (US) to USS to prevent any confusion with the United States (US) in your discussion. 

Second, are the British Household Panel Survey (BHPS) and USS (see above) data based on random samples of households?  If so, you should state that this fact. 

Third, are wages adjusted for inflation across time periods?  If not, please do so.  In the manuscript, you should also state whether wages have been adjusted for inflation.

Lastly, please edit your manuscript for minor grammatical problems.  For instance, you should change "23,764 observation" to "23,764 observations" near the bottom of page 6.  Similarly, change "Our results show that for the population on average," to "Our results show that for the population, on average," on page 9. 

We would appreciate receiving your revised manuscript by Mar 13 2020 11:59PM. To enhance the reproducibility of your results, we recommend that if applicable you deposit your laboratory protocols in protocols.io, where a protocol can be assigned its own identifier (DOI) such that it can be cited independently in the future. For instructions see: http://journals.plos.org/plosone/s/submission-guidelines#loc-laboratory-protocols

We look forward to receiving your revised manuscript.

Kind regards,

Bryan L. Sykes, Ph.D.

Academic Editor

PLOS ONE

Journal Requirements:

Reviewers' comments:

Reviewer's Responses to Questions

**Comments to the Author**

1. Is the manuscript technically sound, and do the data support the conclusions?

Reviewer #1: Yes

2. Has the statistical analysis been performed appropriately and rigorously? 

Reviewer #1: No

3. Have the authors made all data underlying the findings in their manuscript fully available?

Reviewer #1: No

4. Is the manuscript presented in an intelligible fashion and written in standard English?

Reviewer #1: Yes

5. Review Comments to the Author

Reviewer #1: This is an interesting research article. The importance of the topic is highlighted and potentially the findings can have a strong policy impact.

However, the following issues were noted, listed in the order in which they appear in the paper:

Introduction:

- It would be good to know what is considered a high/low intergenerational correlation. Are there any standards in the literature? One can look at the sign of the correlation, but it would be good if for example 0.25 is considered as a high correlation in this domain.

- It is unclear from the introduction which outcomes are going to be looked at and how. The novelty of this study is that it looks at the "intergenerational correlation in health and wages". From this statement, it seems that the correlation studies will be between childhood health and adult wages, which is not the case. Instead, health and wages are two separate outcomes in the analysis. In other words, the correlation is only between generations rather than outcomes. It is a bit confusing since a substantial paragraph is dedicated to the Grossman model emphasising the correlation between health and wages. Maybe you should be a bit clearer from the beginning on what outcome you are going to look at and how you will assess correlation.

Data and methods:

- More information should be given about the surveys. Are they conducted yearly? How many data points do you have?

- How are generations defined? In the first survey, you have one cohort of individuals divided into two policy periods. Given that the same individuals (two generations) will be use for the analysis of period 1991-1999 and 1999-2010, how do you account for the fact that the correlation in each outcome in the second period will be largely influenced by the correlation in the first period?. This is even more problematic since the the last period 2010-2017 is drawn from a separate sample.

- The correlation in each outcome has potentially lots of determinants other than policy landscape. Using the "mean correlation for the sample for each wave" is confusing: how does this remove the effect of genetics and other key determinants?? It seems that the econometric model lacks theoretical grounds.

- Why do you use fixed effects? More explanation should be provided.

- How correlation is measured? At the individual level by pairing each child with parent? This is critically important but not clear enough.

- The unit of analysis of the econometric model is unclear, is it at the individual level? Or is it average correlation? This is unclear. Do you use any clustering to account for family effect?

- How do you take inflation into account?

Results and discussion:

- Be careful you have two figure 1.

- Figures 2a,b and c are very difficult to read.. might be useful to present this in another way.

- you should emphasise in your limitations that measuring health by SAH is not great, since it is subject to lots of biases. Although, it is even more problematic since this bias might not go in the same direction by SES, which can cause problems for the sub-group analysis. Could you use presence chronic disease as a measure of health for example?

- "The findings suggests that the impact of government policy related to reducing inequalities on

the family unit is lower than the impact on individuals." Where is the result supporting this statement?

- the statement "Our findings provide some evidence that the correlation in health and wages does evolve over

time and may be responsive to changes in policy focus especially for those at the highest and

lowest end of the socioeconomic spectrum." does not seem to be fair considering that poor correlation has been found overall.

6. PLOS authors have the option to publish the peer review history of their article (what does this mean?). If published, this will include your full peer review and any attached files.

Reviewer #1: No

---

## [Author Response · Author response to Decision Letter 0]

30 Mar 2020

Please see the word document attached

---

## [Decision Letter · Decision Letter 1]

21 May 2020

PONE-D-19-32818R1

Understanding the role of policy on inequalities in the intergenerational correlation in health and wages: Evidence from the UK from 1991-2017.

PLOS ONE

Dear Dr. Brown,

Thank you for submitting your manuscript to PLOS ONE. After careful consideration, we feel that it has merit but does not fully meet PLOS ONE’s publication criteria as it currently stands. Therefore, we invite you to submit a revised version of the manuscript that addresses the points raised during the review process.

Thank you for undertaking a major revision of your initial submission.  Reviewer 1, however, requests a four  additional clarifications. Although Reviewer 1 has "never seen such an analysis before" using the fixed effects model specification you employ (Point #1), I am familiar with such research in econometrics, specifically research on labor economics and health.  Reviewer 1's other concerns, especially Point #2, are valid; please explain why you employ a fixed-effects model over a difference-in-difference (DD) or a difference-in-difference-in-difference (DDD) approach to estimating the effects of the policy changes.  Please justify your use of estimating the average fixed effect in the population (before and after policy periods) compared to simply estimating the effect of the policy using DD or DDD approaches.

Also, although you have edited your paper considerably, a number of minor typographical errors remain.  For example, in the second paragraph under the methodology section, "In2001" should be "In 2001," and "US" should be "USS".  Please edit your submission again to ensure that all acronyms, spacing, and other minor grammatical errors are fixed. 

We look forward to receiving your revised manuscript.

Kind regards,

Bryan L. Sykes, Ph.D.

Academic Editor

PLOS ONE

Reviewers' comments:

Reviewer's Responses to Questions

**Comments to the Author**

1. If the authors have adequately addressed your comments raised in a previous round of review and you feel that this manuscript is now acceptable for publication, you may indicate that here to bypass the “Comments to the Author” section, enter your conflict of interest statement in the “Confidential to Editor” section, and submit your "Accept" recommendation.

Reviewer #1: All comments have been addressed

2. Is the manuscript technically sound, and do the data support the conclusions?

Reviewer #1: Partly

3. Has the statistical analysis been performed appropriately and rigorously? 

Reviewer #1: I Don't Know

4. Have the authors made all data underlying the findings in their manuscript fully available?

Reviewer #1: Yes

5. Is the manuscript presented in an intelligible fashion and written in standard English?

Reviewer #1: Yes

6. Review Comments to the Author

Reviewer #1: Thank you very much for revising the manuscript. I think this version is much clearer. However, there are still some points which I do not understand fully.

1- My major concern is related to the model used in the analysis. To me, a linear regression with fixed effects is used for panel data where 1) the explanatory variable is a random variable with a normal distribution and 2) where the unit of analysis is the individual. In your case, your explanatory variable is time expressed in year (without a normal distribution) and where the unit of analysis is time as well. Therefore I am questioning the validity of the model in this case. I have never seen such an analysis before so it would be good to have more explanation on this type of analysis.

2- Related to this point, I also don't understand how your model enables to attribute the changes observed to the change in policy landscape. To me the present analysis is purely descriptive and no inference can be made about any causal effect. I think that should be made clear.

3- I still don't understand how the cohorts are defined. From your table 1, it seems that the mean age for each of the 3 time periods are quite similar. If these cohorts are constituted with the same families, how come the age do not change over time? I must have misunderstood something in the construction of the cohorts, so maybe more explanation on this would be useful is the method section.

4- I understand you are looking at the three outcomes separately to see if the correlation goes in the same direction over time. Given the interconnection between health and wages as you nicely described in the introduction, I would find it useful in the discussion to emphasise more on this aspect. The policy implications are a bit light at present.

7. PLOS authors have the option to publish the peer review history of their article (what does this mean?). If published, this will include your full peer review and any attached files.

Reviewer #1: No

---

## [Author Response · Author response to Decision Letter 1]

28 May 2020

Please see the response to reviewer document for changes made to the manuscript.

---

## [Decision Letter · Decision Letter 2]

2 Jun 2020

Understanding the role of policy on inequalities in the intergenerational correlation in health and wages: Evidence from the UK from 1991-2017.

PONE-D-19-32818R2

Dear Dr. Brown,

We are pleased to inform you that your manuscript has been judged scientifically suitable for publication and will be formally accepted for publication once it complies with all outstanding technical requirements.

With kind regards,

Bryan L. Sykes, Ph.D.

Academic Editor

PLOS ONE

Additional Editor Comments (optional):

Reviewers' comments:

Reviewer's Responses to Questions

**Comments to the Author**

1. If the authors have adequately addressed your comments raised in a previous round of review and you feel that this manuscript is now acceptable for publication, you may indicate that here to bypass the “Comments to the Author” section, enter your conflict of interest statement in the “Confidential to Editor” section, and submit your "Accept" recommendation.

Reviewer #1: All comments have been addressed

2. Is the manuscript technically sound, and do the data support the conclusions?

Reviewer #1: Yes

3. Has the statistical analysis been performed appropriately and rigorously? 

Reviewer #1: I Don't Know

4. Have the authors made all data underlying the findings in their manuscript fully available?

Reviewer #1: Yes

5. Is the manuscript presented in an intelligible fashion and written in standard English?

Reviewer #1: Yes

6. Review Comments to the Author

Reviewer #1: Many thanks for your comments and for your work. It is clearer to me now and I think this article will be of value to the research community.

I am still not very convinced by the model, but this may be because I have never seen such a specification. To me a linear model should be used when the explanatory variable is a random and normally distributed variable, which is not the case of "time period".

7. PLOS authors have the option to publish the peer review history of their article (what does this mean?). If published, this will include your full peer review and any attached files.

Reviewer #1: No

---

## [Editor Report · Acceptance letter]

4 Jun 2020

PONE-D-19-32818R2 

Understanding the role of policy on inequalities in the intergenerational correlation in health and wages: Evidence from the UK from 1991-2017. 

Dear Dr. Brown:

I'm pleased to inform you that your manuscript has been deemed suitable for publication in PLOS ONE. Congratulations! Your manuscript is now with our production department. 

Kind regards, 

on behalf of

Dr. Bryan L. Sykes 

Academic Editor

PLOS ONE